# Focus on Pancreatic Cancer Microenvironment

**Fabiana Prattic̀o and Ingrid Garajová ***

Medical Oncology Unit, University Hospital of Parma, 43100 Parma, Italy; fabiana.prattico@unipr.it
* Correspondence: ingegarajova@gmail.com; Tel.: +39-0521702660

**Abstract:** Pancreatic ductal adenocarcinoma remains one of the most lethal solid tumors due to its local aggressiveness and metastatic potential, with a 5-year survival rate of only 13%. A robust connection between pancreatic cancer microenvironment and tumor progression exists, as well as resistance to current anticancer treatments. Pancreatic cancer has a complex tumor microenvironment, characterized by an intricate crosstalk between cancer cells, cancer-associated fibroblasts and immune cells. The complex composition of the tumor microenvironment is also reflected in the diversity of its acellular components, such as the extracellular matrix, cytokines, growth factors and secreted ligands involved in signaling pathways. Desmoplasia, the hallmark of the pancreatic cancer microenvironment, contributes by creating a dense and hypoxic environment that promotes further tumorigenesis, provides innate systemic resistance and suppresses anti-tumor immune invasion. We discuss the complex crosstalk among tumor microenvironment components and explore therapeutic strategies and opportunities in pancreatic cancer research. Better understanding of the tumor microenvironment and its influence on pancreatic cancer progression could lead to potential novel therapeutic options, such as integration of immunotherapy and cytokine-targeted treatments.

**Keywords:** pancreatic ductal adenocarcinoma; tumor microenvironment; desmoplasia; cancer-associated fibroblasts; immune cells; cytokines; cancer therapy; immunotherapy

## 1. Introduction

Pancreatic cancer is the third leading cause of cancer-related death worldwide with an increasing incidence of approximately 1% annually in both sexes. PDAC is the most common histological subtype accounting for around 90% of pancreatic cancer. It is often associated with a poor prognosis due to late diagnosis and limited treatment options, with a 5-year OS rate of only 12–13% [1,2]. PDAC arises mainly from noninvasive precancerous lesions, mostly from microscopic PanIN, less frequently from IPMNs or MCN [3]. In the preneoplastic lesions, cells accumulate genetic and epigenetic alterations in oncogenes and tumor suppressor genes. Most common are alterations of the oncogene KRAS and the tumor suppressor genes CDKN2A, TP53, and SMAD4, triggering aberrant signaling cascades [4]. In addition to the extensive molecular alterations in neoplastic cells, the TME plays a crucial role in pancreatic tumorigenesis, tumor growth and metastasis as well as treatment resistance [5]. This review aims to elucidate the complex interactions within the tumor microenvironment and their therapeutic implications.

## 2. Characteristics of the Pancreatic Cancer Microenvironment

There is a growing interest in a better understanding of the tumor microenvironment, its role in resistance to systemic therapies and its potential as a target for novel therapeutic approaches. PDAC has a particularly prominent mesenchymal compartment within its stroma, which includes fibroblasts, ECM components, immune cells, nerves and endothelial cells. All these components might contribute to tumor progression [6] (Figure 1).

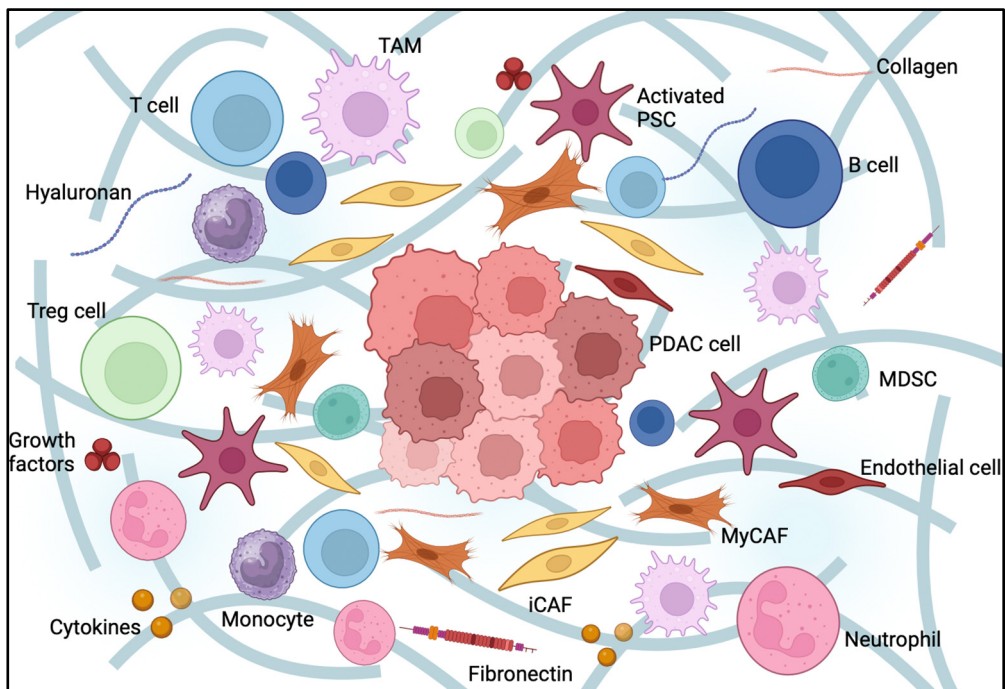

**Figure 1.** Pancreatic cancer tumor microenvironment (TME). Pancreatic ductal adenocarcinoma TME composition. PDA stroma is comprised of cellular and acellular components, such as fibroblasts, myofibroblasts, pancreatic stellate cells, immune cells, blood vessels, extracellular matrix and soluble proteins, such as cytokines and growth factors. Interactions between these elements promote the desmoplastic structure, creating a physical barrier responsible for PDAC treatment resistance.

## 2.1. Desmoplasia

PDAC is characterized by an extensive fibrosis, termed desmoplasia, which promotes tumor progression and inhibits drug penetration and uptake [7]. In desmoplasia, the PSC transforms from a quiescent cell to that of an activated SMA expressing cell via the TGFβ and PGFGF signaling pathway, promoting ECM synthesis and deposition [7]. TGFβ has a dual nature: the tumor-intrinsic TGF-β plays a tumor suppressive function in the early stages through SMAD4-regulated genes, but the stromal-derived TGF-β promotes cancer growth and immunosuppressive mechanisms in the later stages, upon SMAD4 inactivation [8]. Desmoplasia lead to compression of microvasculature and overall hypovascularity in the TME, establishing a hypoxic microenvironment which contributes to pancreatic cancer aggressiveness [9]. In addition, pancreatic cancer cells can secrete antiangiogenic factors, such as angiostatin, endostatin and PEDF, supporting a hypoxic microenvironment [10,11]. Thus, desmoplasia is associated with a poor prognosis and its components might be potential therapeutic targets of pancreatic cancer.

## 2.2. Extracellular Matrix and Structural Proteins

The extracellular matrix consists of proteins, glycoproteins, proteoglycans, polysaccharides and soluble signaling molecules that provide a physical scaffold for its surrounding cells. The ECM can be divided into two matrices: the BM and the IM. The BM supports epithelial and endothelial cells and separates them from the IM, which makes up the main stroma and plays a major role in cell migration, cell adhesion, angiogenesis, tissue development and repair [12]. The major components of the basement membrane are the network-forming collagens, such as type IV and VIII collagen and, in the interstitial matrix, the fibrillar-forming collagens type I, II, III, V, XI, XXIV, XXVII and the beaded filament type VI collagen [13]. Cancer cells secrete MMPs, which physiologically regulate collagen production and assembly [14,15], leading to a complex of pro- and antitumor signals from BM's degradation products [13]. These structural modifications contribute to increased

stiffening of the PDAC tissue, influencing cellular behavior and regulating cell proliferation, differentiation, gene expression, migration, invasion and metastasis [16,17].

### 2.2.1. Hyaluronic Acid

HA is a matrix glycosaminoglycan in PDAC stroma, linked to elevated IFP. Given its large amount, HA has been identified as a promising potential target for therapeutic strategy. Hyaluronidase encapsulated by polyethylene glycol (PEGPH20) has been formulated to improve in vivo half-life and degrade HA. KPC (KrasLSL-G12D/+; Trp53LSL-R172H/+; Cre) mice, a mouse model that recapitulates the human disease, treated with a single cycle of PEGPH20 in addition to the standard chemotherapy with gemcitabine experienced a substantial decrease in tumor volume, suggesting that reducing IFP enhances the tumor perfusion of chemotherapeutic agents [18]. A different study showed that PEGPH20 reduced vascular collapse and increased intratumoral delivery of gemcitabine in vivo, leading to an increase in animal survival compared with chemotherapy alone [19]. However, the administration of PEGPH20 combined with modified FOLFIRINOX (mFOLFIRINOX) resulted associated with a reduced survival in treatment-naïve patients with metastatic PDAC [20]. A phase III trial (HALO 301 study) [21] comparing gemcitabine/nab-paclitaxel with or without PEGPH20 in untreated patients with stage IV PDAC has been unsuccessful. Similarly, Atezolizumab plus PEGPH20 did not improve OS or PFS in previously treated patients with advanced pancreatic cancer [22]. Therefore, the available data on the use of this enzyme did not show a statistically significant clinical benefit in PDAC patients.

### 2.2.2. Fibronectin

FN, a large multidomain glycoprotein dimer assembled by cell-driven forces into a fibrillar array, is another component of TME. It serves as a scaffold for other matrix proteins and as a binding site for soluble factors within the tumor microenvironment. FN regulates proliferation and metastatic behavior of multiple cell types, largely through mechanisms involving integrin-mediated signaling [23]. Its induction during tumorigenesis can sustain proliferative signaling, promote angiogenesis, facilitate invasion and metastasis, modulate growth suppressor activity and regulate anti-tumoral immunity. Moreover, fibronectin plays a key role in the development of gemcitabine resistance through ERK1/2 activation. Thus, combining FN-blocking agents plus gemcitabine-based chemotherapy could potentially overcome chemoresistance in PDAC, providing better clinical outcomes [24].

### *2.3. Cells in Tumor Microenvironment*

PDAC desmoplastic stroma is rich in cancer-associated fibroblasts (CAFs) and infiltrated by immune cells and their soluble functional molecules [7]. Exploring the biological and functional characteristics of TME is of great interest for identifying potential novel therapeutic options.

### 2.3.1. Cancer-Associated Fibroblasts

During tumor progression, CAFs are the main protagonists in dysregulating collagen turnover, leading to excessive collagen deposition and tumor fibrosis [25]. Due to their various origins, CAFs form a heterogeneous cell population with strong differences in morphology, cell–cell interaction and expression profile. CAFs express myofibroblast markers, such as α-SMA, vimentin, type XI collagen, fibronectin, FSP-1 and FAP [26], and promote tumor growth. Genome-wide studies have identified distinct subtypes of CAFs in PDAC: myo-fibroblastic (myCAFs) or inflammatory phenotypes (iCAFs). While the majority of fibroblasts express low levels of αSMA and high levels of FAP, a subpopulation of FAP+ cells show elevated αSMA expression. MyCAFs, characterized by high expression of α-SMA and low expression of IL6, are activated by direct contact between tumor cells and PSCs. They produce extracellular matrix components and promote stromal deposition [27]. iCAFs, with low α-SMA and high IL-6 levels and inflammatory features, are located farther away from tumor cells and activated by paracrine factors secreted from tumor cells. IL-6, IL-11, CXCL1,

CXCL2 and LIF are upregulated in iCAFs, that specifically expressed HAS1 and HAS2, two enzymes responsible for the synthesis of hyaluronic acid [27]. Different pathways are enriched in iCAFs, including IFN-γ response, TNF/NF-κB, IL2/STAT 5 and IL6/JAK/STAT3 in humans. Il-1/LIF/JAK/STAT pathway could activate iCAFs, which in turn suppress the immune response and ECM deposition, accelerating the occurrence and progression of PDAC [28]. Furthermore, CAFs contribute to chemoresistance via various mechanisms. PAI-1, a cytokine produced by CAFs, activates Erk/Akt signaling and suppresses caspase-3 activation [29]. Similarly, interleukin 6 (IL6), also produced by CAFs, contributes to drug resistance mediating CXCR7 expression in tumor cells [30,31]. Especially under hypoxic conditions, CAFs produce high levels of TGFβ, conferring stem cell-like properties to tumor cells and increasing their chemoresistance [32]. Moreover, the complex ECM produced by CAFs interferes with therapeutic response by producing a protective barrier and interacting with integrins and cadherins [33–37]. Considering the heterogeneity of CAFs, understanding their functions might lead to develop specific strategies for each subgroup. It has been shown that chemoresistance driven by CD10+GPR77+CAFs [38] could be mitigated by the depletion of these cells with an anti-GPR77 antibody. Another approach might be reprogramming CAFs to a quiescent phenotype by inhibiting activating pathways, such as NFκb signaling, which has been shown to improve response to cisplatin in ovarian cancer xenografts [31]. Sherman et al. have demonstrated that activated CAFs also express vitamin D receptor. Calcipotriol, a vitamin D analog, can reverse CAFs back to stellate cells, normalize the TME and increase gemcitabine concentration in the tumor, resulting in improved response in PDAC models [39]. In addition, there is an invasive and migratory CAF phenotype associated with ROCK-pathway activation, that can be targeted with the ROCK inhibitor, fasudil. This has shown to reduce collagen deposition, enhance gemcitabine uptake, and improve treatment response in a transgenic PDAC model [40].

### 2.3.2. Pancreatic Stellate Cells

During PDAC, quiescent pancreatic stellate cells are activated and transformed into the myofibroblast-like phenotype and secrete molecules like TGFβ, IL-6, SDF-1, HGF and galectin-1, involved in pancreatic cancer progression [41]. Furthermore, PSCs can also induce desmoplasia via numerous signaling pathways, including IL-6, paracrine SHH signaling, VDR pathway and CXCL12/CXCR4 pathway [42]. The CXCL12/CXCR4 axis plays a critical role in promoting cancer cell proliferation, migration, invasion, angiogenesis and metastasis. Combined therapies with CXCR4 antagonists, such as CXCL14, have shown great promise in preclinical models, but most of the studies investigated drug administration early in the disease course, which does not always reflect clinical reality [43]. Another potential drug target is SHH signaling. Induced by KRAS via NF- kB signaling, pancreatic cancer cells express SHH to activate the target gene GLI1 and create a tumor-supportive microenvironment [44]. SHH signaling activates PSCs into activated myCAFs [45], affecting the ratio of myCAFs/iCAFs in PDAC tumors, and influences the expression of inflammatory signaling molecules from the iCAF population and the immune cell infiltration in TME. Although well-known Hh signaling antagonists, such as vismodegib (GDC-0449), have been investigated, they have not yielded clinically meaningful results [46,47]. Available data suggest that ormeloxifene, a non-hormonal, nonsteroidal oral contraceptive molecule, has a property of inhibiting the SHH pathway and, in combination with gemcitabine, could serve as a novel therapeutic strategy for PDAC [48].

### 2.3.3. Endothelial Cells

Solid tumors are commonly characterized by an abnormal vasculature. In response to hypoxia due to poor vasculature, tumors usually activate mechanisms to stimulate angiogenesis, supporting tumor growth and metastasis. Pancreatic cancer cells and other immunosuppressive cells in the tumor microenvironment induce angiogenesis by secreting pro-angiogenic factors, cytokines and growth factors, such as VEGF, regulated by multiple signaling pathways. STAT3 expression in pancreatic cancer cells activates VEGF expression,

promoting angiogenesis [49]. Similarly, MUC1 induces hypoxia, stimulating VEGF-A and PDGF-B production and contributing to endothelial cell tube formation [50]. Additionally, activated NF-κB in pancreatic cancer cells can upregulate VEGF [51]; this pathway can be suppressed by xanthohumol, with subsequent angiogenesis inhibition [52]. Although therapeutic agents that inhibit angiogenesis or normalize the tumor vasculature are not part of the standard of care in the treatment of PDAC [53], some patients might benefit from such approaches. For example, mice with highly vascular PDAC have shown modest but significant benefit in overall survival with VEGF receptor inhibition [54,55]. Regarding lymphatic vessels, they play a fundamental role not only in transporting tumor antigens to draining lymph nodes for presentation and activation of tumor-reactive T cells by leukocytes, but also because they represent the main dissemination route in PDAC [56,57]. Lymphatic vasculature contribution to PDAC pathogenesis remains poorly understood and is likely to be complex. Chemokines are involved in lymph-angiogenesis and cell migration. For example, CCL21, secreted by lymphatic endothelial cells, attracts dendritic cells to lymph nodes via lymphatics, while tumor cells expressing CCR7 may exploit this mechanism to disseminate to lymph nodes [58–60]. Similarly, CXCL12 produced in lymph nodes may attract CXCR4-expressing cancer cells or leukocytes [61,62]. Thus, tumor-associated lymph-angiogenesis is a critical step for pancreatic cancer progression, especially for lymph node metastasis.

## 3. Neural Invasion in Pancreatic Cancer

NI consists of the presence of cancer cells along nerves and/or within the epineural, perineural and endo-neural spaces [63]. Epineural invasion is characterized by cancer cells that directly touch the epineurium without penetrating it; differently, in perineural and endo-neural invasion, cancer cells invade the perineural and the endo-neural sheet, respectively [63–65]. NI, particularly endo-neural invasion, correlates with increased neuropathic and inflammatory pain, which significantly impacts patient outcomes [65]. There is a reciprocal interaction between tumor and nerve cells that creates a microenvironment favorable to tumor growth. Nerve-secreted substances promote tumor invasion and growth, while tumor-secreted molecular mediators stimulate nerve axons' growth toward tumor cells [66], supporting tumor development and influencing its microenvironment [67]. Additionally, tumor induced-neural sprouting enhances a cancer cell–nerve crosstalk, known as axonogenesis [68]. SCs, the principal glial cells of the peripheral neural system [69], regulate neural growth and pain signaling through the production of a variety of growth factors and pain-related molecules, such as the NGF, the GDNF and the BDNF [70]. Neurotrophin antibodies, including anti-NGF, anti-BDNF, anti-NT-3, and anti-NT-4/5, have shown promising results in preclinical studies [71]. A recent phase III study has demonstrated the efficacy and safety of tanezumab, a monoclonal antibody anti-NGF, in patients with cancer pain predominantly caused by bone metastasis, who were already on opioid therapy. However, the long-term efficacy of anti-NGF therapy beyond 8 weeks remains unclear; therefore, future research should focus on evaluating the extended therapeutic benefits of this treatment [72]. Lastly, understanding the molecular mechanisms behind NI could potentially lead to innovative treatments that improve outcomes and quality of life of patients.

## 4. Immunosuppression

The correlation between anti-tumor immunity and pancreatic cancer progression has been the subject of several studies. The immune system has a dual activity: it prevents the transformation of mutated cells into tumor cells and promotes pancreatic cancer progression by creating favorable conditions for immunosuppression and metastasis [73,74]. Tumor immunity plays an important role in cancer patient prognosis, as evidenced by the presence of CD8+ TILs involved in killing tumor cells [75,76]. Normally, APCs, such as macrophages and DCs, process tumor antigens and display them on MHC I molecules. Consequently, CD8+ T cells are activated to kill tumor cells via the granzyme, perforin, and Fas/FasL (Fas ligand) pathways. On the other side, pancreatic cancer cells promote immunosuppression

by inhibiting CD8+ T cell activation and decrease HLA class I expression, escaping the tumor infiltration by cytotoxic T-cells [77]. Furthermore, pancreatic cancer cells can evade Fas-mediated immune surveillance by expressing a nonfunctional FasR (Fas receptor) to resist Fas-mediated apoptosis and by expressing functional FasL to induce apoptosis in Fas-sensitive activated T-cells [78].

## 5. Immunosuppressive Cells

Tumors have developed strategies to successfully evade the host immune system, utilizing various molecular and cellular mechanisms. In PDAC, MDSCs, TAMs, and Tregs contribute to establishing an immunosuppressive TME [79].

### 5.1. Myeloid-Derived Suppressor Cells (MDSCs)

MDSCs, a heterogeneous population of immature myeloid cells in spleen and tumor, play a critical role in immunosuppression. In the GEMM and orthotopic mouse model of PDAC, myeloid cells, including MDSCs and TAMs, are the most abundant immune cells [80]. MDSCs are characterized by the expression of CD11b and CD33 and peripheral blood MDSCs level may be a predictive biomarker of chemotherapy failure in pancreatic cancer patients [81]. Pancreatic cancer induces the proliferation and mobility of MDSCs from the bone marrow to the tumor microenvironment [82] via cytokines, especially GM–CSF, which promotes the differentiation of myeloid progenitor cells into MDSCs [83]. MDSCs exert their immunosuppressive function through different pathways. For instance, they release ROS, under the influence of cytokines, such as TGFβ and IL-10, inducing oxidative stress in T cells and inhibiting the CD3ζ chain translation [84]. PMN-MDSCs (polymorphonuclear MDSCs) take up, process and present antigens on MHC I proteins. These antigen-MHC I complexes present antigens to CD8+ T cells, inducing immune tolerance and facilitating immune evasion [85]. Furthermore, MDSCs express enzymes. like Arg1 and iNOS, which deplete L-arginine, an essential amino acid involved in T cells' proliferation and activation [86]. The STAT3 signaling pathway, activated by IL-10, not only induces Arg1 expression but also upregulates PD-L1 expression on MDSCs [87], further suppressing T cell activation.

### 5.2. Tumor-Associated Macrophages (TAMs)

Macrophages are phagocytic cells involved in the innate immune system. The PanIN regulates macrophage function via IL-13, polarizing them from M1 to M2 phenotype, which shows immunosuppressive function and promotes tumor growth [88]. In the pancreatic cancer microenvironment, various factors, such as CSF-1, IL-4, IL-13, TGFβ and IL-10, promote myeloid progenitor cell differentiation into monocytes and macrophages [89,90], which in turn secrete immunosuppressive cytokines, chemokines and enzymes, such as TGFβ, IL-10, CCL17, and CCL22 [91]. TAMs express high levels of Arg1, interfering with the metabolism of effector T cells [92], support regulatory Treg recruitment and inhibit CD8+ T cells [93]. Additionally, NLRP3 signaling in macrophages promotes the differentiation of CD4+ T cells into tumor-promoting Th2 cells, Th17 cells and Tregs, while suppressing Th1 cell polarization [94]. TAMs also contribute to the development of desmoplasia. In fact, M2 macrophages promote pancreatic fibrosis [95] and, overall, macrophages drive fibrosis, immunosuppression and metastasis through the PI3Kγ pathway [96]. Lastly, TAMs can stimulate PSC proliferation and ECM secretion via TGFβ1 and PDGF respectively [97], involved in multiple steps of pancreatic cancer progression (Figure 2).

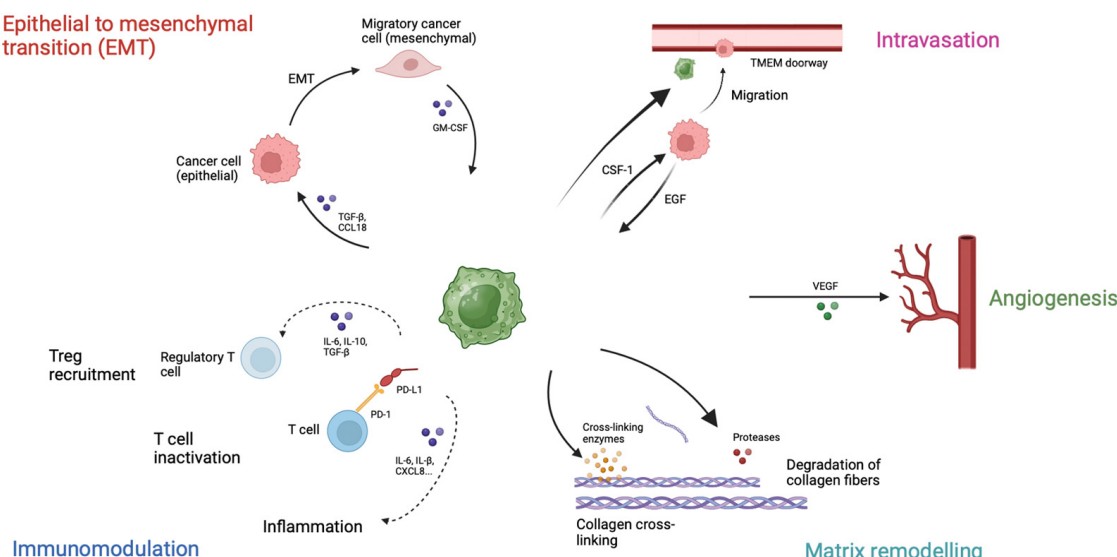

**Figure 2.** Role of Tumor-associated macrophages (TAMs). TAMs inactivate cytotoxic T cells through PD-L1 expression, produce cytokines to recruit regulatory T cells and create an inflammatory environment. These macrophages are involved in extracellular matrix remodeling by producing proteases, such as metalloproteinases, that degrade collagen fibers, and cross-linking enzymes, which contribute to the stiffness of the extracellular matrix. Furthermore, TAMs secrete VEGF and promote angiogenesis, which is involved in tumor progression and metastasis. Tumor-associated macrophages migrate with cancer cells to blood vessels and create TMEM (tumor microenvironment of metastasis) doorways, permitting cancer cells to disseminate in the circulation. Lastly, TAMs play a role in epithelia to mesenchymal transition by producing TGF-β and CCL18, allowing cancer cells to migrate.

### 5.3. Tumor-Associated Neutrophils (TANs)

Only limited information is available on the roles of TANs. In cancer patients, peripheral blood neutrophils level and circulating neutrophil-to-lymphocyte ratio represent biomarkers of poor clinical outcome [98]. Neutrophils are attracted to the TME in response to TNF-α and IL-12 secreted by cancer cells. Via TGF-β signaling, TANs are polarized into the N1 subtype, which inhibits cancer growth by activating CD8+ T cells through releasing TNF-α, CCL-3, CXCL-9, and CXCL-10 [99]. In contrast, Treg-derived IL-35 induces the polarization of neutrophils into the N2 subtype, which has a pro-carcinogenic role due to various secreted molecules [100]. N2 neutrophil-derived ROS, RNS and NE contribute to carcinogenesis and promote cancer progression [101]. In addition, neutrophils produce HGF and MMPs, which facilitate angiogenesis, tumor invasion and metastasis [102,103].

### 5.4. Regulatory T Cells (Tregs)

Tregs, also known as suppressor T cells, are a subtype of T cells that play an important role in maintaining tolerance to self-antigens and preventing autoimmune disease by downregulating or suppressing effector T cells [104]. CD4+ CD25+ Tregs, recruited into the TME by pancreatic cancer cells, perform a significant role in immunosuppression during pancreatic cancer progression. However, the precise mechanism of their immunomodulatory functions in PDAC remains unclear. A high level of Tregs is associated with poor prognosis and low survival rate [105]. Intratumor Tregs express elevated levels of CTLA-4 and PD-1 compared to Tregs in pancreatic lymph nodes (Pan LNs) and peripheral inguinal lymph nodes (iLNs) [106]. The impact of Treg depletion has been investigated in GEMMs, revealing a pro-carcinogenic effect mediated by Tregs interaction with tumor-associated CD11+ DCs [106]. Prolonged binding between intratumor Tregs and CD11+ DCs selectively impacts CD8+ T cell activation, leading to reduced levels of IFN-γ, CD44, and Granzyme B, as well as decreased tumor size and increased overall survival [103]. Conversely, Treg depletion has been found to induce myeloid cell recruitment into the

TME via a CCR1-dependent pathway, resulting in suppressed anti-tumor immunity [107]. Furthermore, Tregs demonstrated plasticity towards Th17 cells (Foxp3+RORγ+), promoting cancer growth and evasion through the production of IL-17, IL-23 and TGF-β [107]. However, an alternative study suggest that Treg depletion induces increased expression of chemo-attractants to myeloid cells (CCL3, CCL6, CCL8) and immune-suppression genes in fibroblasts (PD-L1, Arg1), consequently reprogramming myofibroblasts to an inflammatory phenotype and enhancing their pro-carcinogenic function [108]. Overall, Tregs exhibit complex crosstalk with CAFs and immune compartments within the TME, resulting in both pro- and anti-tumor effects in desmoplastic cancers such as PDAC.

## 6. Secreted Signaling Molecules

In TME, different subpopulations of cells secrete various signaling molecules, typically TGF-β or pro-inflammatory cytokines, altering the tumor microenvironment (Table 1).

**Table 1.** Pro-inflammatory cytokines. Cytokines, low-molecular-weight proteins that regulate many biological processes, play a dual role in cancer: they modulate the tumor microenvironment and directly affect cancer cells. CAF cancer associated fibroblast; DC dendritic cell; EMT epithelial to mesenchymal transition; PSCs pancreatic stellate cells; TAM tumor-associated macrophage; NEU neutrophil. VEGF vascular endothelial growth factor; TGFβ transforming growth factor beta; IL interleukin; TNF tumor necrosis factor; PDAC pancreatic ductal adenocarcinoma.

| Cytokine | Cell Sources | Function in PDAC |
|---|---|---|
| IL-8 | PDAC cell<br>CAF<br>NEU | Acts as an angiogenic factor. Promotes cell growth, survival and tumorigenesis. Contributes to metastasis, by regulating MMP-2. |
| IL-6 | PDAC cell<br>TAM<br>CAF<br>NEU | Promotes oncogenesis through JAK2-STAT3 activation, angiogenesis through VEGF induction, cancer cell migration and EMT. |
| IL-1β | PDAC cell<br>TAM<br>CAF<br>NEU<br>DC | Promotes cancer growth, invasion and metastasis. Recruits proangiogenic macrophages. |
| TNF-α | TAM<br>NEU<br>PSC | Increases PDGF expression, stimulating fibrogenesis. Promotes angiogenesis by inducing VEGF production, cancer cell proliferation and metastasis. |
| IL-10 | TAM<br>Treg<br>M2 macrophage | Involved in tumor proliferation, migration, and proteolytic activity of MMP2 and MMP9. On the other side, immunostimulatory and anti-tumor properties, including the inhibition of angiogenesis and metastasis. |
| TGF-β | CAF<br>M2 macrophage<br>Th2 lymphocyte | Inhibits cell cycle progression in early stage, promotes tumor growth, invasion and metastasis by inducing EMT in advanced stage. |

### 6.1. Interleukin 8 (IL-8)

Interleukin 8 (IL-8), a strong pro-inflammatory cytokine, could serve as a good marker in predicting the prognosis of patients [109] and mainly acts as an angiogenic factor, associated with an increased production of VEGF, VEGFR (VEGF receptor) and NRP-2, which are crucial for angiogenesis in growing pancreatic tumor [109]. It is also involved in the MAPK pathway by enhancing the phosphorylation of ERK, implicated in cell growth, survival and tumorigenesis [110]. The TRAIL can selectively induce apoptosis, engaging its receptors TRAIL-R1 and TRAIL-R2. This leads to increased expression of uPA and IL-8, involved in

invasion, metastasis and tumor progression, especially in cells overexpressing TRAF2 or Bcl-xL. However, TRIAL-receptors are also implicated in non-apoptotic pathways and it was demonstrated that non-apoptotic signaling in PDAC cells is mediated by TRAIL-R1 and enhanced by overexpression of TRAF2 or Bcl-xL [111]. Based on these data, inhibition of Bcl-xL and TRAF2 could represent a targeting strategy, improving the TRAIL-based therapy in the treatment of PDAC [112].

### 6.2. Interleukin 6 (IL-6)

Interleukin 6 (IL-6) is another interleukin involved in inflammation, cell proliferation and differentiation, and its gene polymorphisms are associated with pancreatic cancer. IL-6 activates JAK, a tyrosine kinase, which in turn activates MAPKs, PI3Ks and STATs signaling pathways [113]. IL-6 cooperates with oncogenic K-Ras to activate a detoxification program for ROS via the MAPK/ERK signaling cascade [114]. This detoxification process is particularly important because it enables cancer cells to survive in stressful environments and resist apoptosis. IL-6 activates the gp130 receptor, which subsequently leads to Jak2 activation. Activation of the oncogenic JAK2/STAT3 pathway enhances REG3A expression, accelerates cell cycle progression by increasing CyclinD1 expression and upregulates the anti-apoptosis Bcl-2 family [115]. Importantly, REG3A activation further enhances the JAK2/STAT3 pathway, forming a positive feedback loop that amplifies the oncogenic effects of the IL-6/JAK2/STAT3 pathway, linked to inflammation-related tumorigenesis [116]. Additionally, Jak2 phosphorylates and activates STAT3. Studies have shown that the inhibition of IL-6/STAT3 signaling pathway reduces EMT and inflammation in pancreatic cancer cells [114]. For example, the STAT3 inhibitor NSC74859 opposes PSC-induced migration and EMT-related markers (Snail and cadherin-2) expression in pancreatic cancer cells [115]. In addition, the IL-6/STAT3 signaling pathway can methylate the SOCS3 via DNMT1, leading to pancreatic cancer progression and metastasis. Therefore, targeting STAT3 or DNMT1 might represent a potential strategy in the treatment of pancreatic cancer [117]. For example, embelin, an anti-inflammatory and anti-tumor drug, reduces cell proliferation and induces apoptosis by inhibiting STAT3 pathway and activating p53 signaling [118]. Lastly, combination therapy with IL-6 blockade could modulate immunological features of PDAC and enhance the effectiveness of anti-PD-L1 (anti-programmed death-1-ligand 1) checkpoint inhibitors [119].

### 6.3. Interleukin 1β (IL-1β)

IL-1β, another pro-inflammatory cytokine, involved in tumor growth and metastasis, activates NF-κB, upregulates COX-2 and leads to chemoresistance in pancreatic adenocarcinoma [120,121]. Its expression inhibits components of the integrin signaling pathway, such as vinculin and α5-integrin, via a JNK-dependent mechanism, promoting cells' migratory potential [122]. Specifically, the IL-1β -511CT/-31TC genotype significantly correlates with an elevated risk for pancreatic cancer, particularly in advanced and undifferentiated tumors [123]. IL-1β also recruits proangiogenic macrophages, which explains why using an IL-1β antagonist can reduce tumor inflammation, angiogenesis and growth [124]. Additionally, IL-1R (interleukin 1 receptor) levels are elevated in pancreatic cancer, making this receptor a potential therapeutic target. Sandberg et al. demonstrated that the administration of IL-1β with IL-1RA counteracted IL-1β effects in mouse models, as evidence of IL-1RA inhibitory activity [125,126]. Z-360, a CCK2/gastrin receptor antagonist, in combination with gemcitabine, extended the survival period in a lethal pancreatic cancer xenograft mouse model [127]. Furthermore, a phase Ib/IIa clinical trial reported pain improvement in advanced pancreatic cancer patients treated with Z-360 plus gemcitabine. These findings underscore the potential therapeutic efficacy of targeting IL-1β and associated pathways in PDAC treatment.

### 6.4. Tumor Necrosis Factor-Alpha (TNF-α)

TNF-α, a pro-inflammatory cytokine, belongs to the TNF family, which also includes TNF-β, CD40L (CD40 ligand), TRAIL and FasL. In pancreatic cells, TNF-α activates NF-κB [128,129], inhibiting apoptosis, and increases PDGF expression, which strongly stimulates fibrogenesis during early pancreatic inflammation [130]. At high doses, TNF-α exhibits a cytotoxic effect on tumor cells, mediated by TNF-R1. However, in the pancreatic cancer microenvironment, TNF-α binds to TNF-R2, leading to upregulation of the EGFR and subsequent cancer cell proliferation [131]. TNF-R1 is prevalent in tumor and stromal cells, whereas TNF-R2 is generally present on leukocyte infiltrates. Once released, TNF-α binds to TNF-R1 and targets cells responsible for the production of inflammatory cytokines such as IL-6, IL-8, and IL-10 [132]. Several studies have explored targeting TNF-α for pancreatic cancer treatment. In a phase II study, thalidomide reduced TNF levels and improved weight after eight weeks of treatment compared to placebo [133]. Another phase II trial combined thalidomide with capecitabine in advanced or metastatic pancreatic cancer patients, inhibiting TNF mRNA stability and TNF production. This combination reported a median progression-free survival (PFS) of 2.7 months and a median overall survival (OS) of 6.1 months [134]. However, a recent phase I/II study evaluated the efficacy and tolerability of gemcitabine and etanercept, a TNF-α blocker, in 38 patients with advanced pancreatic cancer, but there was no significant improvement in survival [135]. Overall, although preclinical and early clinical trials have not yet successfully targeted TNF-α in cancer treatment, it remains a promising target for future research.

### 6.5. Interleukin 10 (IL-10)

IL-10 is a homo-dimeric polypeptide chain, expressed by various cells, influencing the expression of cytokines, cell surface molecules and soluble mediators, to activate and sustain immune and inflammatory responses. Generally considered an immunosuppressive cytokine, IL-10 inhibits the production of various cytokines, like IL-1α, IL-1β, IL-6, IL-12, IL-18, TNF, LIF and PAF, by activated monocytes/macrophages [136–138]. Additionally, IL-10 is linked with M2-polarized TAMs and their infiltration intensity correlates with pancreatic cancer prognosis. In pancreatic cancer cells, M2-polarized TAMs promote EMT through TLR4/IL-10 signaling, which in turn enhances proliferation, migration and the proteolytic activity of MMP2 and MMP9 [139]. On the other hand, evidence indicates that IL-10 also possesses immunostimulatory and anti-tumor properties [140], including the inhibition of angiogenesis and matrix metalloproteinase 2, thereby inhibiting metastasis [141]. Clinical studies have shown IL-10 upregulation in pancreatic cancer and its correlation with tumor stage and prognosis [142]. Thus, IL-10 might be a potential biomarker for predicting pancreatic cancer prognosis.

## 7. PDAC Therapeutics and Tumor Microenvironment

Over the past few decades, the effectiveness of PDAC treatments in the clinic has not yet improved significantly. However, the increasing understanding of PDAC biology is paving the way for novel therapeutic strategies that target the components of the TME.

### 7.1. Targeting the Immunosuppressive Mechanism

PDACs are generally considered as immunologically "cold" tumors, indicating that they do not induce active anti-tumor immune responses, except for MSI-high subtype [143,144]. This immunological 'coldness' involves several factors, including an immunosuppressive microenvironment characterized by the expression of PD-L1 and IDangioO1 and a low tumor mutation burden that limits neo-antigenicity [145,146]. Additionally, a dense desmoplastic stroma acts as a physical barrier to immune cell infiltration, playing an important role in the limited responses to immune checkpoint inhibitors with the failure of immunotherapy [147,148]. The components of the immunosuppressive system are potential therapeutic targets. Tregs are one of the most promising candidates and can be selectively depleted administering low doses of cyclophosphamide [149] which, in combination with GVAX,

a GM-CSF gene-transfected tumor cell vaccine, showed an enhanced immune response to PDAC [150]. As with many solid tumors, ICIs have been tested in PDAC treatment, primarily targeting CTLA-4 and PD1/PD-L1, but we only have safety data available, making further evaluation of efficacy necessary. For example, the combination of anti-CD40 monoclonal antibody with gemcitabine, with or without anti-PD-1 antibodies, has shown promising outcomes [151]. Thus, significant research is needed in order to investigate potential remarkable benefits offered by immunotherapy in PDAC, as observed in other types of cancers.

### 7.2. Inhibition of Monocyte/Macrophage Recruitment

There is a significant interest in targeting TAMs due to their role in tumor initiation, invasion and metastasis. Targeting the CCL2/CCR2 axis, which plays a crucial role in recruiting monocytes in various cancer types such as pancreatic cancer [152], offers a promising therapeutic approach to hinder the recruitment of monocytes from the bone marrow into the bloodstream [153]. For example, carlumab, an antibody against CCL2, has been tested in clinical trials [154,155] and using carlumab in association with conventional chemotherapy demonstrated preliminary antitumor activity in patients with advanced cancer [156]. Furthermore, PF-04136309, a CCR2 inhibitor, was shown to reduce CCR-2+/CD14+ monocytes and macrophages in the pre-metastatic liver and primary pancreatic tumor. In a phase 1b trial, PF-04136309 administered in combination with FOLFIRINOX chemotherapy proved to be safe, tolerable and recommended for patients with pancreatic cancer [157].

### 7.3. Inhibition of Macrophage Activation

In macrophage activation, CSF-1/CSF-1R signaling plays a crucial role. The use of CSF-1R inhibitor or CSF-1/CSF-1R neutralizing antibody (e.g., PLX6134/PLX3397) represents one of the most advanced methods for targeting macrophage activation [158,159]. Additionally, combining a CSF-1R signaling antagonist with paclitaxel has been shown to enhance the survival of mice with mammary tumors by slowing primary tumor growth and reducing pulmonary metastasis [160]. In glioblastoma mouse models, inhibition of CSF-1R resulted in significant reduction of tumor size and long-term survival [161,162]. Ries et al. demonstrated that treating animal models with a monoclonal antibody (RG7155) significantly reduced F4/80+ TAMs and increased the CD8+/CD4+ T cell ratio. They also found that administering RG7155 to patients led to a substantial depletion of CSF-1R+CD163+ macrophages in tumor tissues, correlating with clinical responses in patients with diffuse-type giant cell tumor [158]. Furthermore, tyrosine kinase receptor inhibitors regulate the immunosuppressive and tumorigenic functions of TAMs [163,164]. Erlotinib in combination with gemcitabine showed a statistically significant, but not clinically improved, survival in advanced pancreatic cancer [165]. Otherwise, in the LAPO7 trial, which evaluated the administration of chemoradiotherapy after induction chemotherapy in locally advanced pancreatic cancer, no differences in OS and PFS were shown with gemcitabine compared to gemcitabine plus erlotinib [166]. Therefore, the inhibition of CSF-1R in vivo demonstrated a decrease in pancreatic tumor-initiating cells and improved chemotherapy efficacy [80], and the interruption CSF-1/CSF-1R signaling eliminated CD206hi TAMs within pancreatic tumors, reprogramming the remaining macrophages to support antitumor immunity [167]. These findings support the role of CSF-1/CSF-1R signaling as an effective therapeutic target for reprogramming the immunosuppressive microenvironment in PDAC. More studies targeting CSF-2/CSF-2R signaling are needed to assess the potential of immunotherapy in pancreatic cancer treatment.

### 7.4. Cancer Vaccine

PDACs can evade the immune system through various mechanisms, each of which presents an opportunity for therapeutic development. One approach involves addressing the limited cancer antigen presentation by exploring different cancer vaccine strategies,

such as peptide-based, virus-based, DNA-based and cell-based vaccines. Despite positive outcomes in phases I and II, some vaccine strategies, including Telovac, PANVAC and algenpantucel-L, failed in phase III trials [168]. Notably, the GVAX trial, which used a tumor cell vaccine transfected with the granulocyte-macrophage colony-stimulating factor (GM-CSF) gene, demonstrated efficacy in mobilizing effector immune cells into the TME. Consequently, treatment strategies that combine GVAX with other therapies are still being pursued [169]. Therefore, personalized RNA neo-antigen vaccines hold a new hope for more efficient pancreatic cancer treatment due to their ability to activate T cells [170].

## 8. Conclusions

The tumor microenvironment is characterized by high heterogeneity, with cancerous cells, immune cells and CAFs, as well as acellular components. We explored the relationship between cellular populations and acellular factors and the important role it has in tumor growth, metastasis and drug resistance. Pancreatic cancer cells produce signaling molecules, recruiting or activating stromal and immunosuppressive cells, such as Tregs, MDSCs, TAMs, and PSCs, within the microenvironment. Over the past two decades, attention has focused on understanding the complex stromal microenvironment of PDAC and its potential impact on cancer treatment. However, further research is needed to understand the precise mechanisms underlying TME–tumor interactions and improve the efficacy of novel therapeutic strategies. Moreover, future directions will be more focused on multiple approaches, targeting different aspects of tumors, to improve patient outcomes and advance a personalized medicine approach to PDAC.

**Author Contributions:** Conceptualization, I.G. and F.P.; methodology, I.G. and F.P.; resources, I.G. and F.P.; writing—original draft preparation, I.G. and F.P.; writing—review and editing, I.G. and F.P.; supervision, I.G.; funding acquisition, I.G. All authors have read and agreed to the published version of the manuscript.

**Funding:** This research was funded by European Organization for Research and Treatment of Cancer (EORTC RP-2146).

**Conflicts of Interest:** The funders had no role in the design of the study; in the collection, analyses, or interpretation of data; in the writing of the manuscript; or in the decision to publish the results. The authors declare no conflicts of interest.

## Abbreviations

APC, antigen-presenting cell; Arg, arginase; BDNF, brain-derived neurotrophic factor; BM, basement membrane; CAF, cancer-associated fibroblast; CCK, cholecystokinin; CXCL12, CXC chemokine ligand 12; CXCR4, CXC chemokine receptor type 4; DC, dendritic cell; DNMT, DNA Methyltransferase; ECM, extracellular matrix; EGFR, epidermal growth factor receptor; EMT, epithelial-mesenchymal transition; FAP, fibroblast activating protein; FAS, first apoptosis signal; FN, fibronectin; FSP-1, fibroblast-specific protein 1; GDNF, glial cell line-derived neurotrophic factor; GEMM, genetically engineered mouse model; GM-CSF, granulocyte macrophage colony-stimulating factor; HA, Hyaluronic acid; HAS, hyaluronic acid synthase; HGF, hepatocyte growth factor; HLA, human leukocyte antigen; ICI, immune checkpoint inhibitor; IFN-γ, interferon γ; IFP, interstitial fluid pressure; IM, interstitial matrix; iNOS, inducible nitric oxide synthase; IPMN, intraductal papillary mucinous neoplasm; JAK, Janus kinase; LIF, leukemia inhibitory factor; MCN, mucinous cystic neoplasm; MDSC, myeloid-derived suppressor cell; MHC I, major histocompatibility complex I; MMP, metalloproteinase; MUC1, Mucin 1; NE, neutrophil elastase; NF-κB, nuclear factor kappa-B; NGF, nerve growth factor; NI, neural invasion; NRP-2, neuropilin-2; OS, overall survival; PEGPH20, pegylated recombinant human hyaluronidase 20; PSC, pancreatic stellate cell; PanIN, pancreatic intraepithelial neoplasia; PDAC, pancreatic ductal adenocarcinoma; PEDF, pigment epithelium-derived factors; PGFGF, platelet-derived growth factor; RNS, reactive nitrogen species; ROS, reactive oxygen species; SC, Schwann cell; SDF-1, stromal cell-derived factor-1; SHH, sonic hedgehog; SMA, smooth muscle actin; SOCS3, suppressor of cytokine signaling 3; STAT, signal transducer and activator of transcription; TAM, tumor-associated macrophage; TAN, intratumoral neutrophil; TGFβ, transforming

growth factor-beta; Th cell, T helper cell; TIL, tumor-infiltrating lymphocyte; TLR, Toll-like receptor; TME, tumor microenvironment; TNF, tumor necrosis factor; TRAIL, TNF-related apoptosis-inducing ligand; Treg, regulatory T cell; uPA, urokinase-type plasminogen activator; VDR, vitamin D receptor; VEGF, vascular endothelial growth factor.

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
