# Peer review of "Focus on Pancreatic Cancer Microenvironment"

_curroncol, doi:10.3390/curroncol31080316_

Round 1

Reviewer 1 Report

Comments and Suggestions for Authors

This review article serves as a repository of many literatures but does not provide inspiring insights.  Some informations do not keep updated, for example, the authors mentioned the promising results of using PEGPH20 in treating experimental mice (Lines 89-102).  But actually, the clinical trial of PEGPH20 had been failed and terminated in 2019.  Additionally, the authors spent a big space stating CAFs, stellate cells, and immune cells like macrophages, neutrophils, etc.  In recent years, single-cell RNA sequencing has been extensively exploited in the research of tumor microenvironment.  The authors should update the informations obtained from the field. 

Comments on the Quality of English Language

Because of the 53% match by iThenticate report, the text throughout the whole manuscript has better been rewritten and reorganized extensively. 

Author Response

This review article serves as a repository of many literatures but does not provide inspiring insights.  Some informations do not keep updated, for example, the authors mentioned the promising results of using PEGPH20 in treating experimental mice (Lines 89-102).  But actually, the clinical trial of PEGPH20 had been failed and terminated in 2019.  

We agree with this comment; therefore, we modified and clarified this point (lines 100-115). 

Additionally, the authors spent a big space stating CAFs, stellate cells, and immune cells like macrophages, neutrophils, etc. In recent years, single-cell RNA sequencing has been extensively exploited in the research of tumor microenvironment. The authors should update the informations obtained from the field.

Thank you for your suggestions. We described components and interactions within the tumor microenvironment and their therapeutic implications. 

Because of the 53% match by iThenticate report, the text throughout the whole manuscript has better been rewritten and reorganized extensively.

We have made English revision as suggested.

Reviewer 2 Report

Comments and Suggestions for Authors

The authors of this manuscript describe in a comprehensive and understandable manner the various aspects of pancreatic adenocarcinoma microenvironment and its relation to tumor growth and expansion. There is an even presentation of cellular and non-cellular components of PDAC microenvironment and an interesting section with the therapeutic potential of targeting these components. 

I have only two minor comments for the authors. The first has to do with the too many abbreviations inside the text. A small list at the end with the abbreviations would greatly facilitate the inexperienced reader. 

The second refers to the comment in lines 542-544 regarding TKIs. Since erlotinib is an already approved TKI for PDAC treatment, there must also be a comment about it (even if there is no research). 

Author Response

The authors of this manuscript describe in a comprehensive and understandable manner the various aspects of pancreatic adenocarcinoma microenvironment and its relation to tumor growth and expansion. There is an even presentation of cellular and non-cellular components of PDAC microenvironment and an interesting section with the therapeutic potential of targeting these components.

Thank you for your comments.

I have only two minor comments for the authors. The first has to do with the too many abbreviations inside the text. A small list at the end with the abbreviations would greatly facilitate the inexperienced reader. 

Thank you for this suggestion, we added a list with the abbreviations.

The second refers to the comment in lines 542-544 regarding TKIs. Since erlotinib is an already approved TKI for PDAC treatment, there must also be a comment about it (even if there is no research).

We agree with the reviewer and we added a comment (lines 507-513).

Reviewer 3 Report

Comments and Suggestions for Authors

Dear Authors, “Focus on Pancreatic Cancer Microenvironment” is an exciting topic, but the paper has major flaws.

Recently, there have been a lot of studies/ reviews on the pancreatic tumor microenvironment. It is a complex subject, and the Introduction should be very clear. The author should give valid reasons for selecting this topic.

There is no connection between each topic. Each title stands on its own.

Need to improve English.

Figure 1 didn’t explain anything. It needs to be redrawn to show the roles of players in the tumor microenvironment.

The immune part should be very clear. It needs to have two or more figures to explain the pathway involved.

The conclusion part is very confusing. Please give clear perspectives on all the topics covered.

The review article should undergo major revision before being accepted for publication.

Thank you

Comments on the Quality of English Language

The review needs extensive revision for language and grammar.

Author Response

Recently, there have been a lot of studies/ reviews on the pancreatic tumor microenvironment. It is a complex subject, and the Introduction should be very clear. The author should give valid reasons for selecting this topic.

Thank you for your suggestion. We modified the introduction to make this part more clear.

Need to improve English.

We have done English revision, as suggested. 

Figure 1 didn’t explain anything. It needs to be redrawn to show the roles of players in the tumor microenvironment. The immune part should be very clear. It needs to have two or more figures to explain the pathway involved.

Thank you for your suggestions. Figure 1 represents the tumor microenvironment, its heterogeneity and complexity. We added another figure to explain some pathways involved.  

The conclusion part is very confusing. Please give clear perspectives on all the topics covered.

We modified this part to clear all the topics, as suggested.

Round 2

Reviewer 3 Report

Comments and Suggestions for Authors

The authors did an excellent job modifying the manuscript according to the comments. The manuscript can be accepted for publication. Thank you